# Combining array tomography with electron tomography provides insights into leakiness of the blood-brain barrier in mouse cortex

Georg Kislinger[1,2], Gunar Fabig[3], Antonia Wehn[4,5], Lucia Rodriguez[1,2], Hanyi Jiang[1,2,6], Cornelia Niemann[1,2], Andrey S Klymchenko[7], Nikolaus Plesnila[4,8], Thomas Misgeld[1,2,8], Thomas Müller-Reichert[3], Igor Khalin[4,9†], Martina Schifferer[2,8*†]

[1]Institute of Neuronal Cell Biology, Technical University Munich, Munich, Germany; [2]German Center for Neurodegenerative Diseases (DZNE), Munich, Germany; [3]Experimental Center, Faculty of Medicine Carl Gustav Carus, Technische Universität Dresden, Dresden, Germany; [4]Institute for Stroke and Dementia Research (ISD), LMU University Hospital, LMU Munich, Munich, Germany; [5]Department of Neurosurgery, University of Munich Medical Center, Munich, Germany; [6]Department of Psychiatry and Psychotherapy, University Medicine Greifswald, Greifswald, Germany; [7]Laboratoire de Bioimagerie et Pathologies, Université de Strasbourg, Illkirch, France; [8]Munich Cluster of Systems Neurology (SyNergy), Munich, Germany; [9]Normandie University, UNICAEN, INSERM UMR-S U1237, Physiopathology and Imaging of 19 Neurological Disorders (PhIND), GIP Cyceron, Institute Blood and Brain, Caen, France

*For correspondence:
martina.schifferer@dzne.de

†These authors contributed equally to this work

Competing interest: The authors declare that no competing interests exist.

**Abstract** Like other volume electron microscopy approaches, automated tape-collecting ultramicrotomy (ATUM) enables imaging of serial sections deposited on thick plastic tapes by scanning electron microscopy (SEM). ATUM is unique in enabling hierarchical imaging and thus efficient screening for target structures, as needed for correlative light and electron microscopy. However, SEM of sections on tape can only access the section surface, thereby limiting the axial resolution to the typical size of cellular vesicles with an order of magnitude lower than the acquired xy resolution. In contrast, serial-section electron tomography (ET), a transmission electron microscopy-based approach, yields isotropic voxels at full EM resolution, but requires deposition of sections on electron-stable thin and fragile films, thus making screening of large section libraries difficult and prone to section loss. To combine the strength of both approaches, we developed 'ATUM-Tomo, a hybrid method, where sections are first reversibly attached to plastic tape via a dissolvable coating, and after screening detached and transferred to the ET-compatible thin films. As a proof-of-principle, we applied correlative ATUM-Tomo to study ultrastructural features of blood-brain barrier (BBB) leakiness around microthrombi in a mouse model of traumatic brain injury. Microthrombi and associated sites of BBB leakiness were identified by confocal imaging of injected fluorescent and electron-dense nanoparticles, then relocalized by ATUM-SEM, and finally interrogated by correlative ATUM-Tomo. Overall, our new ATUM-Tomo approach will substantially advance ultrastructural analysis of biological phenomena that require cell- and tissue-level contextualization of the finest subcellular textures.

## eLife assessment

The present paper describes an **important** methodological development that combines light (confocal) microscopy with scanning and transmission EM and EM tomography. The method expands the level of structural detail accessible to large-volume EM studies and thus represents an approach to integrate analyses of cellular and sub-cellular structures in biological samples. The study, which provides a **compelling** proof-of-principle, will be of particular value to cell biologists interested in the in-depth interpretation of high-resolution ultrastructural information from sparsely distributed targets - at multiple scales and in diverse biological structures.

## Introduction

Volume electron microscopy (EM) provides high-resolution data of target structures that can be used to render complex biological morphologies in three dimensions (*Peddie et al., 2022*). While volume EM thus provides high spatial resolution, searching for a rare nanometer-sized structure of interest in sections of several hundreds of square micrometers is tedious. To bridge these scales, multimodal approaches like correlative light and electron microscopy (CLEM) (*Scher and Avinoam, 2021*) or micro-computed tomography combined with volume EM *Karreman et al., 2017* have been applied. Among the available volume EM approaches, array tomography techniques (*Wacker and Schroeder, 2013*), such as ATUM (*Kasthuri et al., 2015*), have proven particularly powerful for correlating specific biological structures (*Schifferer et al., 2021*; *Snaidero et al., 2020*). Array tomography techniques require section collection onto a solid substrate, such as coated glass, silicon (*Micheva and Smith, 2007*) or plastic tape, and subsequent serial-section SEM. In ATUM-SEM, a reel-to-reel system transfers the sections from the diamond knife onto tape (*Kubota et al., 2018*). The tape strips are assembled on silicon wafers (*Baena et al., 2019*; *Kislinger et al., 2023*), thus generating tissue libraries that can be repetitively imaged by SEM at different resolution regimes. This separates the correlation task into two phases: (1) low-resolution screening for the region previously imaged by fluorescence microscopy and (2) imaging of re-located structures by high-resolution volume EM. High-resolution imaging by array tomography is, however, currently restricted by physical sectioning, resulting in typical voxels of 3 × 3 × 30–100 nm, where SEM determines the lateral resolution of ≈3 nm. As a consequence, array tomography can reveal fine cellular processes down to ≈40 nm, but potentially misses smaller subcellular structures, such as vesicles or contact sites that can be hidden within a section volume. For higher-resolution imaging, typically transmission electron microscopy (TEM) is applied, with ET being the method of choice for high-resolution imaging of sub-cellular volumes (*West et al., 2011*; *Mastronarde, 2005*; *He and He, 2014*; *Kiewisz et al., 2022*; *Redemann et al., 2017*). In ET, semi-thick sections (200–400 nm) are tilted in the electron beam and Fourier transformed for image reconstruction (*Baumeister et al., 1999*; *Young and Villa, 2023*; *Frank, 2006*), but even with serial semi-thick sections (*Soto et al., 1994*), the overall volume that can be reconstructed and visualized in three dimensions is limited.

So far, ATUM could not be combined with ET because the former necessitates attaching sections on a solid support, while the latter as a projection method, needs sections to be deposited on electron-transparent films. While section deposition is usually irreversible in ATUM, we reasoned that plastic tape coating could enable section removal after SEM and transfer onto ET-compatible films. Thus, selected semi-thick sections could be re-inspected by ET – a novel multimodal approach, which we call 'ATUM-Tomo'. We exemplify the ATUM-Tomo approach by assessing structural determinants of BBB leakiness, a key adaptation of the neurovascular unit that controls substance exchange between blood and central nervous system parenchyma and can be locally disturbed in pathologies, including after ischemia, traumatic brain injury or multiple sclerosis (*Zhang et al., 2019*; *Shi et al., 2020*). We had previously characterized BBB leakage around impact-induced microclots using correlative confocal microscopy and ATUM of fluorescent lipid nanodroplets (LNDs) and gold nanoparticles (*Khalin et al., 2022*). At the BBB, the actual barrier function is determined by distinctive ultrastructural features, mostly within the endothelial layer, including tight junctions, pinocytosis, and vesicular transport (*O'Brown et al., 2019*; *Andreone et al., 2017*; *Nahirney et al., 2016*; *Hirano et al., 1994*). As these structures lie below the 30 nm z-resolution limit of ATUM, they eluded our previous correlative imaging. Here, we demonstrate that triple correlation using confocal microscopy, volume SEM in combination with ET provides an integrated view of the cellular environment as well as the

ultrastructural features of the compromised BBB. Our application shows that ATUM-Tomo in combination with CLEM is a powerful multimodal approach, enabling a bridging of scales from the mm to the nm range.

## Results

### Correlative confocal imaging and ATUM-SEM reveals sites of BBB disruption

Controlled cortical impact in mice is a model for traumatic brain injury that involves BBB damage. To mark sites of vessel injury, we injected LNDs and colloidal gold particles of similar size (30 nm) and surface chemistry (PEGylation) ensuring similar biodistribution. LNDs are highly fluorescent, while gold is electron dense – and both accumulate in microthrombi formed in the impact penumbra within 30–60 min post-injection (*Khalin et al., 2022*), enabling the tracking of microthrombi and BBB leakage sites. These particles were injected systemically 60 min after impact, followed by DyLight 649 lectin as a vessel lumen marker just before perfusion with fixative containing glutaraldehyde, which all fluorophores withstand (*Figure 1A*). We sliced the fixed brains in 80 µm thick coronal vibratome sections (*Figure 1A*) to allow confocal imaging and preserve some longer stretches of blood vessels, while restricting the axial depth of the search volume for correlation. After post-fixation in glutaraldehyde, sections were temporarily mounted on a glass slide with a cover slip and recorded ≈2 × 2 mm confocal overview of all cortical layers bearing the lesion and peri-lesional areas (*Figure 1B*). For correlative EM, we identified areas of vascular occlusions indicated by LNDs accumulations and sites of extravasation marked by perivascular LND fluorescence and locally acquired higher magnification confocal image stacks (*Figure 1C*). Tissue preservation was ensured by imaging for a maximum of 1 hr, followed by gentle removal of the cover slip by floating-off in the buffer.

In order to keep the section flat during EM processing and to minimize the search volume, we dissected the smallest possible tissue block around the region of interest based on tissue fiducials as a guide. The tissue was processed for EM by applying a standard reduced-osmium-thiocarbohydrazide-osmium (rOTO) *en bloc* staining protocol (*Hua et al., 2015*) in a way that facilitates the axial and lateral relocation of the area of interest (*Kislinger et al., 2020a*). Production of serial thin (100 nm) sections and collection of these sections on either carbon-coated Kapton (ccKapton) (*Kasthuri et al., 2015*) or carbon nanotube (*Kubota et al., 2018*) tape generated a library of ribbons. A typical 1000-section library was collected, which could then be imaged hierarchically and repetitively. For an overview to facilitate correlation, we imaged the entire area of every 10th section (i.e. 1000 nm axial sampling) with 200 × 200 nm lateral sampling. This allowed us to efficiently screen such a serial-section library for regions of interest, even if the confocal imaging and ultramicrotomy planes were not perfectly aligned.

Regions with microthrombi and extravasation were identified by correlation and selected for high-resolution (4 × 4 × 100 nm) data acquisition by volume SEM. Microthrombi were characterized by a high LND signal, while sites of extravasation showed a perivascular halo. The microclot illustrated here consisted of luminal erythrocyte stalls (*Figure 1D and G*) in combination with platelets and immune cells (*Figure 1D and H*). An accumulation of platelets, interspersed with filamentous, electron-dense fibrin was found in the center and at the ends, 'plugging' the occlusion (*Figure 1D, F and H*). Correlating to the fluorescence pattern, we detected gold particles and brighter but less discrete particles of the same size, presumably representing LNDs, between luminal cells (*Figure 1D*). Healthy endothelium was characterized by an intact glycocalyx and hence lectin staining in the absence of LND fluorescence. In contrast, at sites of extravasation indicated by an LND halo, we found a neutrophil undergoing diapedesis (*Figure 1E*) and abluminal gold particles in its proximity (*Figure 1F*). Thus, we can assume that some microthrombi could serve not only as entry points for nanoparticles, but also for cells from the blood. Within the microthrombus periphery, regions of the endothelium were swollen with intact tight junctions, while in the core, regions with almost no discrete endothelial layer prevailed (*Figure 1H*). Colloidal gold was found at pathologically thinned endothelium, basement membrane, and in pericytes (*Figure 1F–H*).

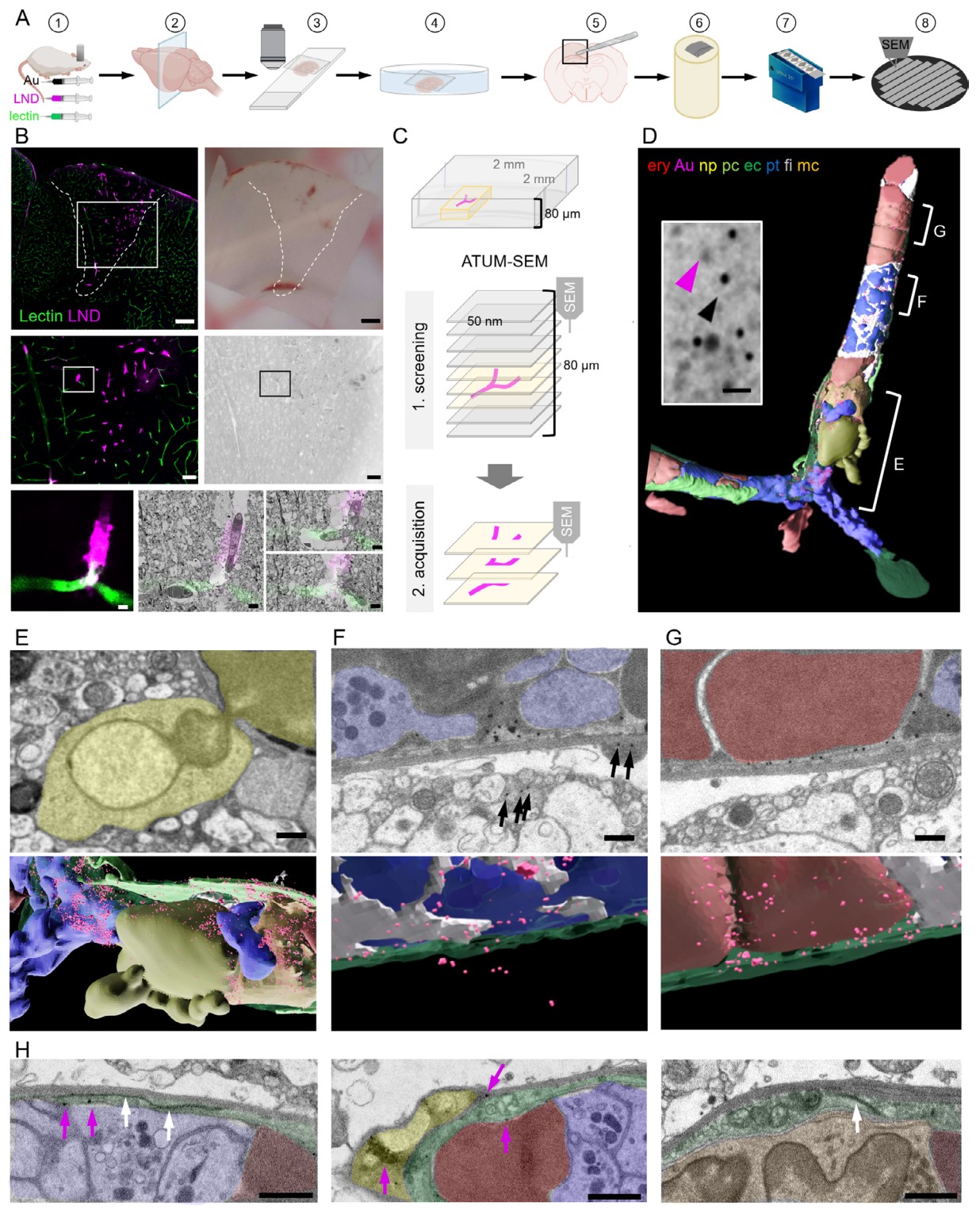

**Figure 1.** Correlative confocal microscopy and automated tape-collecting ultramicrotomy (ATUM)-scanning electron microscopy (SEM) reveal hallmarks of vascular occlusions and extravasation. (**A**) Schematic drawing of the ATUM-SEM correlative light and electron microscopy (CLEM) workflow. Controlled cortical impact (middle) is followed by systemic injection of 30 nm diameter lipid nanodroplets (LNDs) (magenta) and 30 nm diameter colloidal gold nanoparticles (black). DyLight 649 lectin (green) was injected 5 min before perfusion (1). After fixation, coronal vibratome sections are generated (2) and positioned onto glass slides with only laying the cover slip on top of the section for confocal imaging (3). The vibratome section is recovered by immersion into the petri dish (4). The previously imaged region of interest is dissected (5) and processed for EM including embedding

*Figure 1 continued on next page*

*Figure 1 continued*

into resin and contrast enhancement (6). Serial ultramicrotomy and tape collection (7) is followed by wafer mounting and SEM imaging (8). (**B**) Top: Sum projection of a confocal tile scan (lectin, green; LNDs, magenta; left) and corresponding binocular image (right) of the dissected vibratome section. The lesion area is indicated by a dashed line. Scale bars 200 µm. Middle: Lesion area identified in the overview tile scan (box in top image) is relocated in the sum projection confocal image and in the serial section low resolution SEM (right). Scale bars 50 µm. The region of interest (ROI, box) is chosen in the confocal image and re-located in the sum projection SEM image. Bottom: Region of interest confocal image (left) and three single SEM micrographs overlaid with the correlated confocal images (right). Scale bars 5 µm. (**C**) Scheme of the ATUM-SEM strategy for CLEM. A blood vessel of interest (magenta) in an 80 µm thick vibratome section is relocated by screening serial ultrathin sections at low resolution. The target region is reimaged at high resolution (up to 5 × 5 × 50 nm). (**D**) The correlated region in (**B**) was segmented and reconstructed from the ultrastructural data. Endothelium (ec, dark green), LNDs and gold particles (Au, magenta), monocyte (mc, orange), neutrophil (np, yellow), platelet (pt, blue), fibrin (fi, gray), erythrocytes (ery, red), pericyte (pc, bright green) are shown. Inset: SEM image of vessel lumen filled with equally sized, putative LNDs (magenta arrowhead) and gold particles (black arrowhead) and bigger aggregates thereof. Scale bar 100 nm. (**E–G**) Segmented SEM image (top) and three-dimensional rendering thereof (bottom), showing (**E**) the extravasation site from of a neutrophil (yellow), (**F**) extraluminal gold particles (arrows) next to vessel lumen clotted with platelets (blue) and fibrin (gray) and (**G**) stalled erythrocytes (red) interspersed with colloidal gold particles. Scale bars 1 µm. (**H**) Endothelial morphologies from left to right: normal endothelium (dark green) with tight junction (white arrows) and gold particles (magenta arrows); thinned endothelium covered by a pericyte (bright green), swollen endothelium with mitochondria and tight junction. Immune cells (orange), erythrocytes (red), platelets (blue). Scale bars 1 µm.

## Development of reversible section attachment on coated tape for ATUM-Tomo

ATUM-SEM revealed the cellular composition of vascular occlusions and ultrastructural features of BBB leakiness. While transcellular or paracellular pathways are supposed to be involved, so far, early transport mechanisms have not been fully understood. The existence of morphologically normal tight junctions visualized by ATUM-SEM suggests that vesicular transport might be involved. However, resolving gold-particle loaded vesicles at 20–40 nm diameter or tight junctions challenges the axial resolution of ATUM-SEM (≈30 nm). Therefore, we sought to combine ATUM-SEM and ET, as the latter would provide the required resolution (*Michel, 2012*; *Feng et al., 2002*; *Wagner et al., 2012*), but the former would allow efficient screening across an expanded field of view (*Peddie et al., 2022*). For this purpose, we devised a new method to reversibly attach semi-thick ATUM sections to plastic tape for SEM and then transfer them to TEM grids for ET.

We first tested if conventionally collected sections of mouse cortical tissue could be removed from ccKapton or carbon nanotube tape. However, such sections were stuck to the tape despite treatment with organic solvents and any other chemical or physical treatments that we tested (*Figure 2A*). We then hypothesized that adding a thin, dissolvable coating between tape and section could allow gentle detachment (*Figure 2B and C*). Polyvinyl formal (Formvar) was a good candidate, as thin films of this highly solvable polymer are commonly used to support sections for TEM. Moreover, it has been shown that sections can be transferred from Formvar-coated glass coverslips onto TEM grids by hydrofluoric acid (*Paez-Segala et al., 2015*). Therefore, we tested if an ultrathin tissue section could be removed from Formvar-coated tape by organic solvents. Short ccKapton tape strips were Formvar-coated using a standard film casting device (c**f**cKapton), onto which 100–300 nm tissue sections were collected from the diamond knife water bath. Indeed, such sections could be recovered from the tape by incubation in chloroform or 1,2-dichlorethan. To coat longer stretches of tape, we designed a reel-to-reel coating device with a spraying unit (*Figure 2—figure supplement 1*). We mounted such longer cfcKapton tape with tissue sections onto wafers, as required for ATUM-SEM, and subjected them to SEM imaging. We found that Formvar coating induced charging, depending on the thickness of the coat (*Figure 2D*). In addition, imaging at high resolution (<10 nm) prevented subsequent section detachment (*Figure 2—figure supplement 2*).

In order to avoid these problems, we considered other materials that are soluble in organic solvents, and settled on the simple and readily available solution of permanent markers. Marker pens consist of genuine pigmented alcohol-based ink and humectants while a special resin ensures that it is waterproof. We labeled ccKapton with a thin film of permanent marker with a wide tip (cpcKapton) and found that sections attached to such tape could be detached by intense rinsing with acetone. The resulting floating sections could be transferred to a water bath in a drop of acetone using a single-use Pasteur pipette. For longer stretches of tape as needed for serial sections, we rebuilt the coating unit by installing a permanent marker instead of the spraying unit (*Figure 2E*).

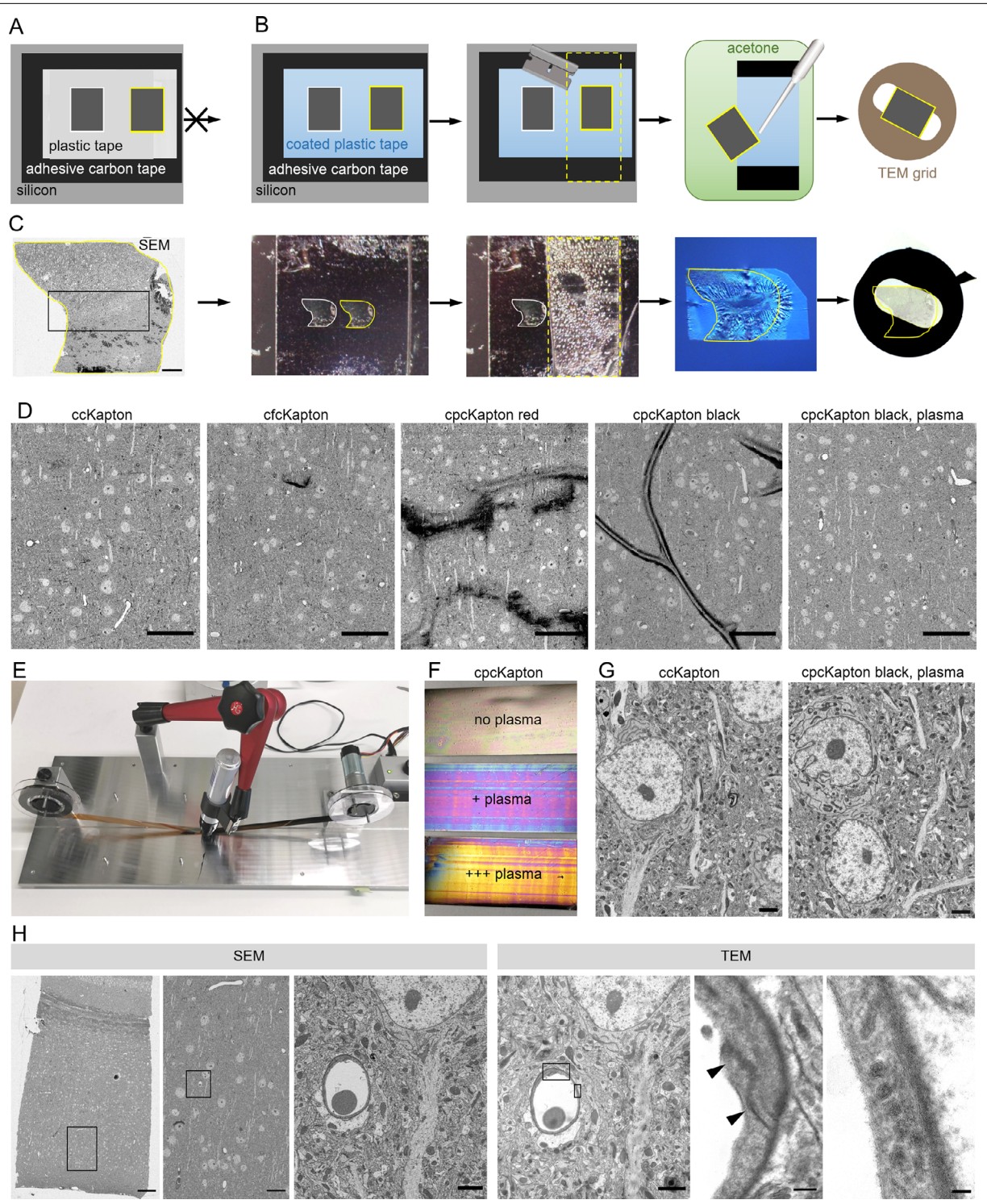

**Figure 2.** Principle of reversible section attachment on coated tape. (**A**) Schematic showing that sections on ccKapton mounted on a silicon wafer cannot be recovered for transmission electron microscopy (TEM). (**B**) Schematics of section removal from cpcKapton or cfcKapton. From left to right: The adhesive tape around the selected section (yellow border) is excised by a razor blade (cutting line: yellow dashed lines) and detached from the wafer. The section is detached from the tape by acetone rinsing using a pipette. The section is collected from the acetone bath onto a TEM slot grid. (**C**) Detachment workflow of a particular selected section (yellow border). From left to right: scanning electron microscopy (SEM) overview image of a section (scale bar 200 nm); the imaged section mounted on the wafer; the section is removed together with the underlying adhesive tape (cutting line: yellow dashed lines); the section is floating on a water bath of a diamond knife after acetone treatment; slot grid with the section. (**D**) SEM images of mouse cortex specimens collected onto different Kapton tapes. From left to right: ccKapton; cfcKapton showing a charging artifact (arrow); cpcKapton

*Figure 2 continued on next page*

*Figure 2 continued*

red with major charging; cpcKapton black without plasma treatment showing section folds; cpcKapton black after intense plasma treatment. Scale bars 50 µm. (**E**) Photograph of the coating unit with a permanent pen and halfway labeled ccKapton. (**F**) Photograph of cpcKapton before discharging (top), after mild (middle) and extensive (bottom) plasma discharging. (**G**) Ultrastructural quality of a cortical tissue section on ccKapton (left) and cpcKapton after plasma treatment (right) imaged at 10 × 10 nm resolution. Scale bars 2 µm. (**H**) Recovery of an ultrathin section. From left to right: SEM overview image of cortex with corpus callosum (scale bar 100 µm). SEM medium resolution image thereof (scale bar 20 µm). SEM high-resolution image (10×10 nm) of a blood vessel cross-section (scale bar 2 µm). TEM image of the same section showing the selected blood vessel (scale bar 2 µm). High magnification image of a tight junction (arrowheads), scale bar 200 nm. High-magnification images of endothelial vesicles (scale bar 50 nm). Black boxes indicate location of the image to the right.

The online version of this article includes the following figure supplement(s) for figure 2:

**Figure supplement 1.** Formvar coating for reversible section collection.

**Figure supplement 2.** Optimizing scanning electron microscopy (SEM) imaging conditions for section detachment.

**Figure supplement 3.** Contamination of section surface after detachment.

**Figure supplement 4.** Estimation of distortion between scanning electron microscopy (SEM) and transmission electron microscopy (TEM) images.

**Figure supplement 5.** Resin test for section recovery.

**Figure supplement 6.** Tape plasma discharging unit construction and use.

Next, we assessed the SEM imaging characteristics of tissue sections mounted on cpcKapton attached to silicon wafers. Coatings with permanent marker of red color, but not black color partially detached from the plastic support and lead to strong charging during SEM imaging (*Figure 2D*). Tissue sections attached to cpcKapton, using pens of either color, showed major tissue section folds (*Figure 2D*). In order to remove these folds, we plasma discharged black cpcKapton tape extensively (see Material and Methods) until the film interference color changed from black to magenta (mild discharging) to gold (extensive discharging) (*Figure 2F*). Such plasma-discharged cpcKapton enabled reversible section attachment at certain SEM imaging conditions (10 nm lateral resolution at 3 µs dwell time) with ultrastructural quality comparable to ccKapton (*Figure 2G*).

We then assessed the tissue ultrastructure in the TEM and the possibility to relocate regions previously imaged by SEM. For this purpose, 80 nm sections on cpcKapton were wafer-mounted, imaged by SEM, and transferred to TEM grids. Despite extensive acetone rinsing the sections stayed intact with well-preserved ultrastructure. The *en bloc* embedding provided strong TEM image contrast without further post staining. If the section area exceeds the slot size, we deposited the region of interest at the grid center. We could efficiently relocate the previously SEM-imaged region in the TEM by using tissue-intrinsic fiducials at low resolution (*Figure 2H*). However, we detected dark deposits (*Figure 2—figure supplement 3A*) on some regions of the section, which probably originate from tissue contact with the coating and accordingly are only superficial. Because ET mainly visualizes the inner tissue areas of the semi-thick sections, we considered these deposits as unproblematic, and cpcKapton hence as a potentially suitable support material for combining ATUM-SEM and ET to bridge across scales (*Figure 3A and B*). We further applied this approach to assess BBB integrity after traumatic brain injury.

## ATUM-CLEM and ATUM-Tomo of nanoparticles at the leaky BBB

To this end, confocal imaging and EM processing of a traumatic brain injury lesion were performed as described for correlative ATUM-SEM. We collected serial 400 nm semi-thick sections of the lesion area covering the entire volume of the vibratome section on cpcKapton. Typically, this resulted in 250 sections that can be mounted onto one or two wafers, dramatically speeding up the correlation process compared to ATUM-SEM (1000 sections on four wafers). We imaged the entire section areas at 100 nm lateral resolution to screen for rough anatomical landmarks. Two vessels with high LND fluorescence signal (*Figure 3C*) were correlated and selected for imaging at higher resolution (20 × 20 nm). Based on this ultrastructural volume and in order to unambiguously identify the localization of nanoparticles, we acquired single images of these target regions at 10 nm lateral resolution. Alignment and three-dimensional rendering of the volume EM data (*Figure 3D*) provided an overview of the underlying neurovascular unit despite the coarser axial resolution compared to ATUM-SEM. Based on these reconstructions and inspection of the ultrastructural volume, we selected sections with gold particles potentially crossing the BBB. These tissue sections were detached by acetone treatment and

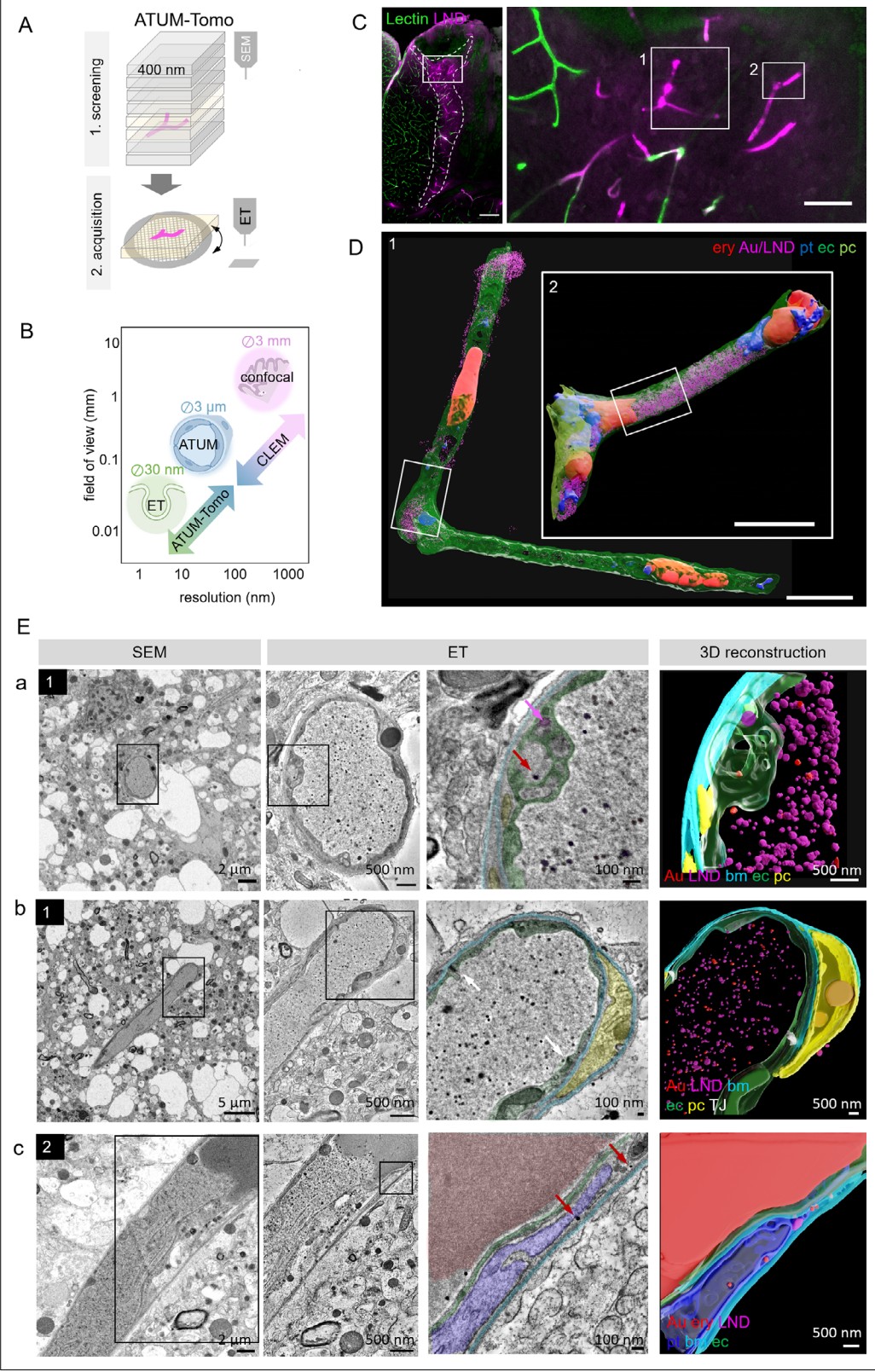

**Figure 3.** Correlative automated tape-collecting ultramicrotomy (ATUM)-Tomo of vascular occlusions. (**A**) Schematic of the ATUM-Tomo workflow. Semi-thick sections are serially imaged by scanning electron microscopy (SEM). Targeted sections (yellow) bearing the site of interest (blood vessel, magenta) are selected for electron tomography (ET). (**B**) Schematic highlighting how ATUM-Tomo and correlative light and electron

*Figure 3 continued on next page*

*Figure 3 continued*

microscopy (CLEM)-ATUM-Tomo bridge scales in resolution and field of view. Transmission electron microscopy (TEM) tomography enables high-resolution imaging e.g., vesicular structures (diameter ~30 nm), ATUM can visualize the cellular composition and morphology (e.g. a neurovascular unit) and confocal provides a large field of view (e.g. highlighting a lesion site in cortex, 3 × 3 mm). (**C**) Confocal microscopy images of the traumatic brain injury lesion. Lectin (green), lipid nanodroplets (LNDs) (magenta). Numbered boxes indicate positions selected for SEM imaging. Scale bars 200 μm, 20 μm (**D**) Reconstruction of selected vessels as indicated in (**C**). Scale bar 10 μm. (**E**) Regions from the vessels 1 (**a, b**) and 2 (**c**) in (**D**) selected for ET. SEM image (first column), corresponding ET low- (second column) and high-magnification images with segmentation (third column) and three-dimensional reconstruction thereof (fourth column). (**a**) Example of particles in endothelial swellings (red arrow gold particle, magenta arrow LND). (**b**) Intact tight junctions (white arrow). (**c**) Endothelial disruption and gold particle localization in a platelet. Endothelium (green), pericyte (yellow), platelet (blue), basement membrane (cyan), LNDs (magenta), gold nanoparticles (red), tight junctions (white).

The online version of this article includes the following video(s) for figure 3:

**Figure 3—video 1.** Rotation of the 3D model as shown in *Figure 3Ea*.

https://elifesciences.org/articles/90565/figures#fig3video1

**Figure 3—video 2.** Rotation of the 3D model as shown in *Figure 3Eb*.

https://elifesciences.org/articles/90565/figures#fig3video2

**Figure 3—video 3.** Rotation of the 3D model as shown in *Figure 3Ec*.

https://elifesciences.org/articles/90565/figures#fig3video3

---

collected onto Formvar-coated slot grids, as needed for subsequent ET. We found that colloidal gold fiducials (20 nm diameter) needed for image reconstruction could be attached to the surface of these sections despite previous SEM imaging. Tomographic reconstructions of semi-thick sections removed from cpcKapton showed surface contamination only at the side that was facing the permanent marker coat (*Figure 2—figure supplement 3B*). The superficial contamination caused reconstruction artifacts in the back projection down to down to a depth of up to 50 nm into the section. Sections that were collected at a nominal cutting thickness of 400 nm showed an actual thickness of 260–300 nm after the tomographic reconstruction. This collapse is due to a mass loss after the interaction of the electron beam with the specimen (*Luther et al., 1988*; *McEwen and Marko, 1998*; *O'Toole et al., 2020*). Moreover, we detected variable lateral shrinkage of about 20% and minor distortions between the SEM and TEM images of ultrathin sections, probably due to beam damage and acetone treatment (*Figure 2—figure supplement 4*). The sections could be recovered even four months after collection and nine months after SEM imaging. Besides Epon/Araldite resin (LX112), we tested the detachment and re-imaging procedure with tissue sections embedded in standard Epon and Durcupan resins (*Figure 2—figure supplement 5*). For all three types of resins, we were able to image by SEM, detach sections and collect them onto grids for re-imaging by TEM tomography.

Despite the above-mentioned minor collapse of the section, the features of the targeted microthrombi and the surrounding neurovascular unit familiar from conventional thin-sectioning ATUM-SEM could be readily discerned: The unit's cellular layers with endothelial cells, basement membrane and pericytic processes could be visualized, even at pathological sites. Moreover, the injected colloidal gold particles and LNDs were accumulated in the vessel lumen, with the identification of LNDs — compared to SEM images—actually facilitated in the ET by the increased resolution (*Figure 3E*; *Figure 3—videos 1–3*). We observed that the endothelial layer showed local edema and detachment from the basement membrane, while tight junctions were still intact (*Figure 3E*). Gold particles were detected in platelet vesicles, vacuoles of endothelial cell swellings, as well as in areas of endothelial detachment from the basement membrane (*Figure 3E*). We also recovered and reconstructed tomograms from nine serial semi-thick sections (*Figure 4*). Comparing the SEM reconstruction with the one from the corresponding serial ETs (*Figure 4A*) clearly showed that SEM imaging visualizes only the section surface resulting in a coarse three-dimensional rendering, while ET provided information of the entire section thickness (*Figure 4B–D*; *Figure 4—videos 1–3*). Taken together, this correlative ATUM-Tomo approach enabled the targeted visualization of extravasating nanoparticles and their ultrastructural organelle carriers within reconstructed CNS neuropil as contextualized by confocal and volume SEM imaging. With our valuable approach, we bridged micro- to nanoscales for the examination of healthy and diseased tissue samples by electron microscopy.

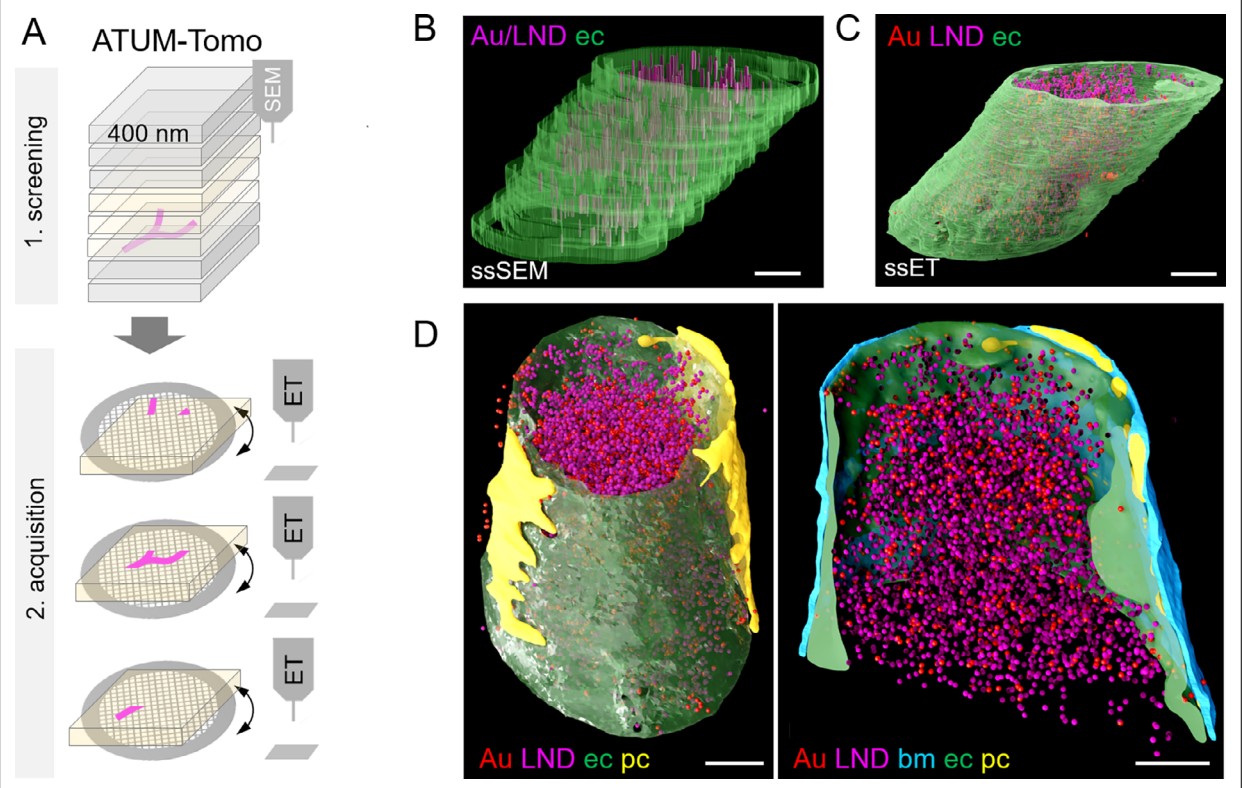

**Figure 4.** Correlative automated tape-collecting ultramicrotomy (ATUM)-Tomo with serial-section electron tomography (ET). (**A**) Schematic of the ATUM-Tomo workflow. Semi-thick sections are serially imaged by scanning electron microscopy (SEM). Targeted sections (yellow) bearing the site of interest (blood vessel, magenta) are selected for serial section ET. Tomograms from consecutive ET volumes can be aligned. (**B–D**) The boxed region in *Figure 3D* (number 2) has been subjected to ET including nine consecutive sections. endothelium in green, pericyte in yellow, platelet in blue, basement membrane in cyan, lipid nanodroplets (LNDs) in magenta, gold nanoparticles in red. (**B**) Non-smoothed, raw reconstructions of nine consecutive SEM images. (**C**) Corresponding region of nine detached sections subjected to ET and reconstructed. (**D**) Side view (left) and longitudinal cut view (right) of the smoothed serial-section ET reconstruction. Scale bars 1 μm.

The online version of this article includes the following video(s) for figure 4:

**Figure 4—video 1.** Rotation of the 3D model as shown in *Figure 4B*.
https://elifesciences.org/articles/90565/figures#fig4video1

**Figure 4—video 2.** Rotation of the 3D model as shown in *Figure 4C*.
https://elifesciences.org/articles/90565/figures#fig4video2

**Figure 4—video 3.** Rotation of the 3D model as shown in *Figure 4D*.
https://elifesciences.org/articles/90565/figures#fig4video3

## Discussion

We and others have shown previously that non-destructive array tomography-type volume EM methods like ATUM-SEM are particularly powerful for correlative workflows, because tissue libraries with re-imaging capabilities are generated (*Schifferer et al., 2021*). In contrast, destructive block-face volume EM methods only allow a single shot acquisition and are limited in field of view. The ATUM imaging process can be separated into screening at low resolution for the sites of interest and subsequent high-resolution volume acquisition. ATUM-SEM allows the rendering of cellular and organellar morphologies at the ultrastructural level. However, the visualization of the detailed subcellular environment of nanoscale structures is limited to the maximal lateral resolution achievable by SEM and the axial resolution set by physical section thickness (30 nm). To increase the axial resolution of target structures, we previously combined ATUM with Focused Ion Beam (FIB)-SEM. ATUM-FIB takes advantage of thick section (2–10 μm) imaging for screening combined with FIB-SEM acquisition of the target region at isotropic voxels (*Kislinger et al., 2020a*; *Kislinger et al., 2020b*). It enables targeted volume imaging at 3 × 3 × 3 nm resolution. The only room temperature volume EM methods that can provide

higher resolution are serial-section TEM or ET (*Peddie et al., 2022*; *Schneider et al., 2021*). While serial-section ET has proven suitable to resolve and reconstruct volumes of abundant (*Soto et al., 1994*; *Rachel et al., 2016*; *Horstmann et al., 2012*) or relatively small biological targets such as spindles (*Kiewisz et al., 2022*; *Redemann et al., 2017*), the visualization of rare or correlated structures is tedious and time-consuming. So far, an automated way of serial TEM can only be done by serial ultrathin sections collected onto grid tape, a plastic tape with holes covered with a thin support film at defined distances (*Graham et al., 2019*). Besides the high costs for the tape, the ultramicrotome has to be automatized to make sure that the sections are positioned onto the slot and not the grid bar. Moreover, a standard TEM has to be re-designed for a reel-to-reel sample loading system (*Maniates-Selvin et al., 2020*; *Yin et al., 2020*), which is not compatible with ET and obviates its application in a standard EM facility.

Our novel ATUM-Tomo approach provides a simple and inexpensive way of combining large-volume SEM with targeted TEM at isotropic, high-resolution voxels. As TEM is a projection method, it requires the removal of the solid substrate, which is needed for SEM imaging. Here, we developed a simple tape coating technique that allows consecutive section inspection by the two EM modalities. Formvar coating resulted in reversible section collection but heterogeneity in film thickness led to charging artifacts, background signal, and sites of irreversible section attachment. We found that marker pen coating provided a simple, reproducible, and homogenous coating. Using pen-coated tape, we were able to image hierarchically up to a lateral resolution of 10 nm by SEM, while retaining the recovery option for transfer onto grids. Detachment of semi-thick sections by acetone did not result in major section distortion. The detachment of ultrathin sections proved more difficult due to section folding and fragility during acetone rinsing. Moreover, surface contamination might potentially cover sites of interest which has more pronounced consequences for the information content of ultrathin than semi-thick sections. Surprisingly, previous SEM imaging does not affect the ultrastructural quality of the ET images. Section folds usually occur at sites within the tissue that have not been imaged by SEM before. This might be caused by reinforced section attachment to the tape caused by prolonged exposure to the electron beam. This effect is even beneficial as it guides the positioning of tissue sections onto the TEM grid slot if it exceeds its area. The SEM volume reconstructed from serial semi-thick sections shows a coarser morphology compared to serial ultrathin sections but still provides the information needed for the selection of sections to be subjected to ET. There was minor tissue loss between consecutive tomograms. This was probably caused by a combination of standard material loss due to sectioning (*Hayworth et al., 2015*; *Kislinger et al., 2020b*) and the incomplete tomographic reconstruction (missing cone) (*Barth et al., 1988*; *Ding et al., 2019*; *Figure 2—figure supplement 3*; *Figure 4B*). While we observed tissue loss because of surface contamination and sectioning, we were able to recover and reconstruct a volume of up to nine consecutive semi-thick sections by ET. Thus, we demonstrate that *en bloc* stained tissue can be sequentially imaged by volume SEM and subsequent ET providing high ultrastructural details.

We inspected microthrombi in traumatic brain injury by both correlative ATUM and ATUM-Tomo. In both cases, CLEM was facilitated by the preservation of lectin and LND fluorescence despite glutaraldehyde fixation, a prerequisite for optimal ultrastructural preservation. While LNDs enable the detection of microthrombi and sites of BBB leakage by confocal microscopy, their size and contrast show variability in SEM and TEM micrographs. Here, the detection of abluminal gold particles served as a clear indicator of extravasation by virtue of the high electron density and discrete size of the gold particles. Nanoparticles were found among luminal cells and close to sites of immune cell diapedesis. By virtue of the novel ATUM-Tomo approach, we captured nanoparticles at sites of endothelial swellings. In line with our results, earlier studies already described endothelial edema preceding FITC-albumin early after middle cerebral artery occlusion (*Krueger et al., 2015*; *Krueger et al., 2019*), another model system for studying BBB leakiness. At later stages, these authors observed loss of endothelial cell integrity with discontinuous plasma membranes and detachment from the basement membrane. However, the DAB staining of FITC-albumin applied in these studies masks the endothelial ultrastructure and thus provides compromised EM information (*Krueger et al., 2019*). In contrast, our CLEM approach provides high ultrastructural quality by chemical fixation with glutaraldehyde. Endothelial swellings were identified at regions without lectin staining, only showing LND fluorescence. Lectin serves as a marker for intact glycocalyx and its loss indicates early pathological events leading to activation of transcellular transport. It was shown that the glycocalyx is critical for BBB

integrity by suppressing caveolin1-dependent endothelial transcytosis following ischemic stroke (*Zhu et al., 2022*). This is in line with the detection of intact tight junctions and gold particles in vesicles at sites of BBB leakiness in the current and previous studies (*Krueger et al., 2015*). In contrast, the paracellular pathway seems not to be the first loophole and gains importance starting 48 hr after the injury (*Knowland et al., 2014*).

In summary, we developed a simple and low-cost method for multimodal EM imaging that combines the strengths of both volume SEM and ET. Our novel ATUM-Tomo approach enables the consecutive inspection of selected areas of interest by correlative serial SEM and TEM, optionally in combination with CLEM. ATUM-Tomo and also correlative ATUM-Tomo can bridge several scales, which is particularly important for ultrastructural analysis of focal pathologies or rare cellular and subcellular events. Both imaging modalities are non-destructive, thus allowing re-imaging and hierarchical imaging at the SEM and TEM levels, which is especially important for precious samples including human biopsies and complex CLEM experiments. While we demonstrate a neuropathology-related application, further biological targets that require high-resolution isotropic voxels and spatial orientation within a larger ultrastructural context can potentially be studied by ATUM-Tomo. This includes the detection of gap junctions for connectomics or for the study of long-range projections (*Holler et al., 2021*) and the subcellular location of virus particles (*Wu et al., 2022*; *Roingeard, 2008*; *Pelchen-Matthews and Marsh, 2007*). Thus, ATUM-Tomo opens up new avenues for multimodal volume EM imaging of diverse biological research areas.

## Materials and methods

### Controlled cortical impact model of traumatic brain injury

All animal experiments were approved by the Ethical Review Board of the Government of Upper Bavaria (ref. number Vet_02-18-39). Animals (male 23–25 g, 8 weeks old) were supplied from Charles River Laboratories (Sulzfeld, Germany). The data were collected in accordance with the ARRIVE guidelines (*Kilkenny et al., 2010*) recommendations (*Guillen, 2012*). Animal husbandry, health screens, and hygiene management checks were performed in accordance with the Federation of European Laboratory Animal Science Associations (FELASA) guidelines and recommendations. 12 C57Bl/6 N male mice, 23–25 g, underwent controlled cortical impact as described above and our in previous studies (*Wehn et al., 2021*).

This model induces a highly reproducible lesion that shares many characteristics of human traumatic brain injury. In short, after induction of anesthesia with buprenorphine (100 mg × kg − 1 Bw) and isoflurane (4%, 30 s), animals were sedated with 1.5–2.5% isoflurane in oxygen/air under continuous monitoring of body temperature and heart rate. After right parietal craniotomy, the impact is directly applied to the intact dura with a pressure-controlled custom made controlled cortical impact device (L. Kopacz, University of Mainz, Germany; 6 ms−1, 0.5 mm penetration depth, 150 ms contact time). After re-fixation of the skull plate and surgical closure, animals are ventilated with 100% oxygen until they re-gained consciousness, then were kept in an incubator at 34 °C and 60% humidity in order to prevent hypothermia. Perfusion, confocal imaging, and correlative ATUM-SEM were performed as previously described (*Khalin et al., 2022*).

### Lipid nanodroplets

Dye-loaded LNDs were produced by spontaneous nanoemulsification by adapting previously described protocols (*Bouchaala et al., 2016*). Briefly, the 1% solution of R18/TPB (prepared as described earlier *Reisch et al., 2014*) in LabrafacWL was mixed with Kolliphor ELP, and the mixture was homogenized under magnetic stirring at 37 °C for 10 min. The amounts of LabrafacWL and Kolliphor ELP were 40 and 60 mg, respectively. Finally, LNDs were obtained by the addition of 230 mg Milli-Q water. Hydrodynamic diameter was measured by dynamic light scattering (DLS) using a Zetasizer Nano ZSP (Malvern Instruments S.A., Worcestershire, UK), based on volume statistics. The size of LNDs was 35.66±1.66 nm, PDI index 0.113±0.002.

### Injection

For injection of lectin and NPs, we used an established femoral arterial catheterization protocol (*Khalin et al., 2020*). 1 mg/ml DyLight 649 Labeled Lycopersicon Esculentum (Tomato) Lectin (Vector

Laboratories, Burlingame, CA, US) for labeling of blood vessels was injected in a quantity of 0.1 ml per mouse. Commercially available 30 nm colloidal gold nanoparticles with density 1.00 g/cm³, methyl terminated, PEG 2000 coated (Sigma-Aldrich, St. Louis, Missouri, USA) were injected through the femoral catheter into the mouse in the same dose as LNDs (4 µL/g animal) (*Khalin et al., 2022*).

### Tissue collection

Animals were transcardially perfused with 4% PFA and 2.5% glutaraldehyde in PBS pH 7.4 in deep anesthesia. The brain was collected and the lesion area of fixed brains was subsequently vibratome-sectioned in the rostral-caudal direction into 80–100 µm sections using a vibratome as previously described (*Ghosh et al., 2015*). In order to ensure high ultrastructural preservation, sections were postfixed for 16–24 hr in the same fixative mentioned above, washed in buffer, and carefully transferred using brushes. Instead of standard mounting which could harm the tissue, sections were mounted onto a glass slide using mounting medium (Aqueous Mounting Medium (ab128982), Abcam, Cambridge, UK) diluted with PBS at a ratio of 1:5. A cover slip (0.17 µm, Menzel Gläser, Braunschweig, Germany) was then placed onto it for imaging. After performing confocal microscopy, the mounted sections were submerged in PBS and the cover slip was carefully removed. Sections could then be transferred back into 24-well plates containing fresh buffer for further processing.

### Confocal imaging for correlation

Imaging was performed using confocal microscopy (ZEISS LSM 900, Carl Zeiss Microscopy GmbH, Jena Germany). Confocal imaging revealed that lectin and LND fluorescence was possible with minimal autofluorescence despite perfusion fixation with glutaraldehyde. A 10 x objective (EC Plan-Neofluar 10 x/0.30 M27) was used with an image matrix of 1024 × 1024 pixels, a pixel scaling of 0.2 × 0.2 µm, and a color depth of 8 bit. The scanned region of approximately 2×2 mm covered the whole cortical thickness of the lesion and peri-lesional area in order to generate overview maps for later correlation. Whole brain images were collected in z-stacks as tile scans with a slice-distance of 5 µm and a total range of 50 µm. Specific regions of interest including the microthrombi and sites of extravasation were collected in the single plane using 25 x magnification (objective: EC Plan-Neofluar 25 x/0.8 Imm Korr DIC M27) with an image matrix of 1024 × 1024 pixel, a pixel scaling of 0.2 × 0.2 µm and a depth of 8 bit. After imaging for a maximal 1 hr, the glass slide was immersed into a petri dish filled with 1 x PBS. As soon as the tissue section was floating from the glass sandwich, it was transferred to a fresh buffer for storage.

### Embedding of vibratome sections for electron microscopy

The sections were stored in PBS at 4 °C for up to one week until the start of the post-embedding. For correlation, the confocal imaging plane was ideally parallel to the sectioning plane during ultramicrotomy. The searchable volume in the axial dimension is predetermined by and restricted to the vibratome section thickness. Usually, heavy metal impregnation leads to the bending of the section. In order to avoid this sample preparation artifact and minimize the lateral searchable area, we dissected a region of approximately 1.5 × 2 mm capturing the lesion and the perilesional area under the binocular using fine scalpels. The fluorescence overview tile scan was used as a template to relocate the previously imaged region. The overall section shape, including big blood vessels and bleedings that represent non-fluorescent regions in the confocal template, were used for this coarse lateral re-location.

We applied a standard reduced osmium thiocarbohydrazide osmium (rOTO) *en bloc* staining protocol (*Kislinger et al., 2020a*; *Kislinger et al., 2023*) including post-fixation in 2% osmium tetroxide (EMS), 1.5% potassium ferrocyanide (Sigma) in 0.1 M sodium cacodylate (Science Services) buffer (pH 7.4). Staining was enhanced by reaction with 1% thiocarbohydrazide (Sigma) for 45 min at 40 °C. The tissue was washed in water and incubated in 2% aqueous osmium tetroxide, washed and further contrasted by overnight incubation in 1% aqueous uranyl acetate at 4 °C and 2 hr at 50 °C. Samples were dehydrated in a graded series of ethanol and infiltrated with LX112 (LADD) resin. For the resin test (*Figure 2—figure supplement 5*), we followed the same protocol while infiltrating samples in Durcupan (11.4 g component A, 10 g component B, 0.3 g component C and 0.1 g component D (Sigma-Aldrich, USA)) and Epon (21.4 g glycidether 100, 14.4 g DDSA, 11.3 g MNA and 0.84 g DMP-30 (Serva, Heidelberg, Germany)), respectively.

## Plastic tape plasma discharging

Plasma discharging increases hydrophilicity and thereby prevents wrinkle formation during serial section collection for ATUM and ATUM-Tomo approaches. In addition, ATUM-Tomo requires several rounds of plasma treatment after marker pen coating as the marker introduces hydrophobicity.

For this end, we built a plasma treatment instrument for tape reels based on a standard instrument (easiGlow, PELCO) with a custom-made vacuum chamber according to the plans proposed by Jeff Lichtman (*Schalek et al., 2011*) and Mark Terasaki (*Baena et al., 2019*), with slight modifications regarding the motor placement and the type of motor (*Figure 2—figure supplement 6*), see also *Kislinger et al., 2023*. We used a dynamic O-ring shaft seal (Kurt J. Lesker Company) and a gear motor with a 150:1 reduction and a suitable controller (Pololu Corporation, Las Vegas, USA) and fixed it directly to the enclosure. In this setting, the motor achieves slow tape speeds (at 12 rpm) while maintaining necessary pulling forces. A more detailed description of the construction and the use as well as a parts list are deposited here: https://github.com/georgkislinger/GKislinger_elife_ATUM-Tomo, copy archived at *Kislinger, 2024*.

We plasma discharged (negative polarity, 0.35 mBar, 15 mA) as many times as needed to achieve a golden instead of a black appearance (*Figure 2F*; *Figure 2—figure supplement 6*). In addition, hydrophobicity was tested by estimating the water contact angle by placing a drop of water onto the tape. It is ready to use if it spreads well with a contact angle below 45° (see also *Kislinger et al., 2023*). This tape can be used for up to one month with one round of plasma discharging before each collection process. However, additional rounds of plasma discharging may increase charging artifacts and impede section recovery as the conductive carbon and pen coatings become thinner.

## Plastic tape marker coating

Initially, we coated ~10 cm long carbon-coated Kapton (ccKapton) (kindly provided by Jeff Lichtman and Richard Schalek) or carbon nanotube (Science Services) tape strips with 1% (w/v) Formvar in chloroform (Sigma) using a standard film casting device for grids (Science Services). In order to generate a homogenous film on longer ccKapton tape pieces, we designed a spray-coating device. This custom-built coating machine was equipped with a holder for the spraying unit (Lubrimat L60, Steidle) and two tape reels driven by a motor (*Figure 2—figure supplement 1*). However, this method proved to be unreliable, as the chloroform caused deterioration of the spraying unit over time which, in turn, the poststain (i.e. lead) to an uneven and contaminated coating. Heterogeneous coating resulted in charging during or detachment problems after SEM imaging.

In order to enable section removal, we marked ccKapton tape on its coated side with a permanent marker (edding 3000, black, edding International GmbH, Germany). We used the before-mentioned coating machine and replaced the spraying unit with a permanent parker. Thereby, the marker was constantly pressed onto the tape as it was pulled under the marker tip to ensure uniform tape coating (*Figure 2E*).

## Section collection onto coated tape

For the volume analysis, the block was trimmed at a depth of 100 µm and a 45° angle on four sides using a trimming machine (EM TRIM, Leica). Serial sections were taken with a 35° ultra-maxi diamond knife (Diatome) at a nominal cutting thickness of 80–100 nm for ATUM-SEM and the TEM tests and 400 nm for ATUM-Tomo at the ATUMtome (Powertome, RMC). For ATUM-SEM, sections were collected onto freshly plasma-treated carbon nanotube tape (Science Services); for ATUM-Tomo, sections were collected on either cpcKapton (carbon- and permanent marker coated) or cfcKapton (carbon- and Formvar-coated) tape. Covering the whole sample thickness of 80 µm and considering its unevenness after contrasting, either 1000 sections at 100 nm thickness (ATUM-SEM) or 400 sections at 400 nm thickness (ATUM-Tomo) were taken. Plastic tape stripes were assembled onto adhesive carbon tape (Science Services) attached to four-inch silicon wafers (Siegert Wafer) and grounded by adhesive carbon tape strips (Science Services). For each sample, 4–6 (ATUM-SEM) or 1–2 (ATUM-Tomo) wafers each bearing 200–300 sections were generated.

## Serial-section acquisition by SEM

ATUM-SEM was performed on a Crossbeam Gemini 340 SEM (Zeiss) with a four-quadrant backscatter detector at 8 kV. Hierarchical imaging captures large areas at low resolution that can, in turn, be

screened for structures of interest that are subsequently reimaged at higher resolution and so forth. In ATLAS5 Array Tomography (Fibics), wafer overview images were generated (1000 nm/pixel). We acquired whole section low-resolution images (0.1 × 0.1 × 1 µm) of every 10th section (thus simulating the thickness of one confocal plane covering 1 µm thickness) to generate overviews that can be correlated with the confocal imaging data. We used blood vessel patterns, bleedings, 1 and the different tissue layers as landmarks. Target regions were identified and acquired at medium resolution (20 × 20 × 200 nm). Within this volume, clots were selected for high-resolution imaging (4 – 20 × 4 – 20 × 200 nm).

For ATUM-Tomo, electron micrographs were acquired in a similar hierarchical scheme on an Apreo S2 SEM (Thermo Fisher Scientific) with the T1 detector. Serial-section acquisition was performed using the Maps2 (Thermo Fisher Scientific) software. We acquired images of whole sections at low resolution (100 × 100 × 400 nm) to facilitate re-location. Anatomical landmarks and vasculature patterns of single SEM overview images and a summed (in Fiji) stack of sections were compared to the confocal data sets in Fiji. Within this volume, we selected a region with clots, and transition zones, and extravasation for high-resolution imaging at 20 × 20 × 400 nm. We acquired subsets of these regions on each or every second section at 10 × 10 × 400 nm or 10 × 10 × 800 nm resolution, respectively.

## Correlation and volume image analysis

There were two levels of correlation: first, we dissected the tissue according to the field of view of confocal imaging, and second, we identified occlusions of interest in the low-resolution SEM data for high-resolution imaging. At each step, anatomical landmarks (bleedings with accumulation of erythrocytes, the section outline, etc.) and vasculature patterns of single and summed sections were compared to the confocal data sets using either Fiji (*Schindelin et al., 2012*) or the ec-CLEM ICY plugin (*Paul-Gilloteaux et al., 2017*). We used both a single SEM image and the sum (in Fiji) of a stack of SEM images for relocation of the occlusions of interest. Serial-section data were aligned by a sequence of automatic and manual processing steps in Fiji TrakEM2 (*Kislinger et al., 2023*). The VAST software was used for manual segmentation of blood vessels (*Berger et al., 2018*) that were rendered in Blender (*Brito, 2018*) for better visualization.

## Section retrieval from coated tape onto TEM grids

Based on SEM-imaging results and the corresponding three-dimensional SEM data, we selected sections of interest for subsequent TEM imaging. We cut the carbon adhesive tape around the selected section on the wafer using a scalpel and peeled off the section-containing piece of cpcKapton from the adhesive tape (*Figure 2B and C*). Next, we submerged the piece of tape containing one single section of interest in a glass petri dish containing acetone. We then created a steady acetone flow with a plastic transfer pipette until the coating between the tape and section was dissolved, resulting in a section floating in the acetone bath. In our experience, the required rinsing time increases with previous beam exposure of the section in the SEM. Once floating in the acetone bath, we collected the single section in a plastic transfer pipette and released it into a water-filled boat (as used for sectioning with a diamond knife, kindly provided by Diatome). Through the surface tension of the water, the section stretched and floated on the water's surface. However, through the detachment from the tape in the acetone bath, it could happen that the sections were physically flipped on the water bath relative to the sectioning direction.

## TEM of ultrathin sections

Ultrathin sections were removed from either cfcKapton or cpcKapton tape using chloroform or acetone. Retrieved sections were collected onto Formvar-coated copper grids (Science Services). Sections were imaged using a JEM-1400+ (JEOL) transmission electron microscope (TEM) equipped with an XF416 (TVIPS) camera operated by the EM-Menu software (TVIPS).

## Electron tomography of semi-thin sections

For ET, single serial semi-thick (400 nm) sections were released as described above and collected on slot grids covered with a Formvar layer of golden interference color. Colloidal gold particles (diameter 20 nm, BBI) serving as fiducials for tomographic reconstruction were then attached to the sections by submerging the grids into a droplet of the pure colloidal gold solution for 1 min. Samples were

blotted to remove residual solution and air-dried. Semi-thick sections were imaged using a Tecnai F30 transmission electron microscope (Thermo Fisher Scientific) operated at 300 kV and equipped with a 4k × 4k CMOS camera (OneView, Gatan) as previously described (*Kiewisz et al., 2022*). Using a dual-axis specimen holder (Type 2040, Fishione, USA), tilt series were recorded from –60° to +60° with 1° increments at a magnification of 4700 x and a pixel size of 2.572 nm applying the SerialEM software package (*Mastronarde, 2003*; *Mastronarde, 2005*). For dual-tilt electron tomography, the grid was rotated for 90° in the XY-plane and the second tilt series was acquired using identical microscope settings (*Mastronarde, 1997*). Both tomographic image stacks of the same positions were reconstructed, combined and flattened using IMOD (*Kremer et al., 1996*; *Mastronarde and Held, 2017*). Serial semi-thick sections were stitched, and combined by using the ZIB Amira (Zuse Institute Berlin, Germany) software package (*Stalling et al., 2005*; *Lindow et al., 2021*). After converting the image data into TIFF format, the image registration was manually refined in Fiji TrakEM2 followed by a final elastic alignment. Segmentation of cellular features was performed manually with the program VAST (*Berger et al., 2018*) and subsequently rendered in Blender (*Brito, 2018*). All movies were generated using Amira.

## Materials and availability statement

Confocal, array tomography, and electron tomography data are available on Bioimage Archive S-BIAD 1274 from August 15, 2024 on.

## Acknowledgements

This work was supported by the DFG under Germany's Excellence Strategy within the framework of the Munich Cluster for Systems Neurology (SyNergy; EXC 2145 – ID 390857198), TRR 274/1 2020 (projects Z01 and B03 – ID 408885537) and FOR Immunostroke (Mi 694/9–1 A03 – ID 428663564). Research in the Müller-Reichert lab was funded by the DFG (grant MU 1423/8–2 and 8–3 to TMR). All animal experiments were supported by DFG grant 457586042. We thank Felix Beyer and Gero Finck (TUM, SyNergy NeuroCore), Dr. Tobias Fürstenhaupt (EM facility at MPI-CBG, Dresden) for excellent technical assistance, and Théo Cerciat (TUM) for segmentation.

## Additional information

### Funding

| Funder | Grant reference number | Author |
| --- | --- | --- |
| Deutsche Forschungsgemeinschaft | SyNergy; EXC 2145 - ID 390857198 | Cornelia Niemann Nikolaus Plesnila Thomas Misgeld Martina Schifferer |
| Deutsche Forschungsgemeinschaft | TRR274 - ID 408885537 | Georg Kislinger Thomas Misgeld Martina Schifferer |
| Deutsche Forschungsgemeinschaft | FOR Immunostroke Mi 694/9-1 A03 - ID 428663564 | Nikolaus Plesnila Thomas Misgeld Igor Khalin |
| Deutsche Forschungsgemeinschaft | MU 1423/8-2 and 8-3 | Thomas Müller-Reichert |
| Deutsche Forschungsgemeinschaft | 457586042 | Nikolaus Plesnila Igor Khalin |

The funders had no role in study design, data collection and interpretation, or the decision to submit the work for publication.

### Author contributions

Georg Kislinger, Conceptualization, Data curation, Investigation, Visualization, Methodology, Writing – review and editing; Gunar Fabig, Data curation, Investigation, Visualization, Methodology, Writing

– review and editing; Antonia Wehn, Investigation, Visualization; Lucia Rodriguez, Hanyi Jiang, Visualization; Cornelia Niemann, Investigation, Methodology; Andrey S Klymchenko, Resources, Methodology; Nikolaus Plesnila, Resources, Supervision, Funding acquisition; Thomas Misgeld, Thomas Müller-Reichert, Resources, Supervision, Funding acquisition, Writing – review and editing; Igor Khalin, Conceptualization, Data curation, Supervision, Investigation, Visualization, Methodology, Writing – review and editing; Martina Schifferer, Conceptualization, Funding acquisition, Investigation, Visualization, Writing – original draft, Project administration, Writing – review and editing

### Author ORCIDs
Gunar Fabig ⓘ https://orcid.org/0000-0003-3017-0978
Nikolaus Plesnila ⓘ https://orcid.org/0000-0001-8832-228X
Thomas Misgeld ⓘ https://orcid.org/0000-0001-9875-6794
Thomas Müller-Reichert ⓘ https://orcid.org/0000-0003-0203-1436
Martina Schifferer ⓘ https://orcid.org/0000-0002-0500-8218

### Ethics

All animal experiments were approved by the Ethical Review Board of the Government of Upper Bavaria (ref. number Vet_02-18-39). Animals (male 23-25g, 8 weeks old) were supplied from Charles River Laboratories (Sulzfeld, Germany). The data were collected in accordance with the ARRIVE guidelines (Kilkenny et al., 2010) recommendations (Guillen, 2012). Animal husbandry, health screens, and hygiene management checks were performed in accordance with Federation of European Laboratory Animal Science Associations (FELASA) guidelines and recommendations. 12 C57Bl/6N male mice, 23-25 g, underwent controlled cortical impact as described above and our in previous studies (Wehn et al., 2021).

Reviewer #1 (Public Review): https://doi.org/10.7554/eLife.90565.3.sa1
Reviewer #2 (Public Review): https://doi.org/10.7554/eLife.90565.3.sa2
Author response https://doi.org/10.7554/eLife.90565.3.sa3

## Additional files

### Supplementary files
• MDAR checklist

### Data availability

Imaging data (confocal images, array tomography and electron tomography data) will be accessible via Bioimage Archive at accession number S-BIAD 1274.

The following dataset was generated:

| Author(s) | Year | Dataset title | Dataset URL | Database and Identifier |
|---|---|---|---|---|
| Georg K, Gunar F, Antonia W, Lucia R, Hanyi J, Cornelia N, Andrey S, Nikolaus P, Thomas M, Thomas M-R, Reichert I, Khalin I, Martina S | 2024 | ATUM-Tomo: A multi-scale approach to cellular ultrastructure by combined volume scanning electron microscopy and electron tomography | https://www.ebi.ac.uk/biostudies/bioimages/studies/S-BIAD1274 | Bioimage Archive, S-BIAD1274 |

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
