## [Editor Report · eLife assessment]

The present paper describes an **important** methodological development that combines light (confocal) microscopy with scanning and transmission EM and EM tomography. The method expands the level of structural detail accessible to large-volume EM studies and thus represents an approach to integrate analyses of cellular and sub-cellular structures in biological samples. The study, which provides a **compelling** proof-of-principle, will be of particular value to cell biologists interested in the in-depth interpretation of high-resolution ultrastructural information from sparsely distributed targets - at multiple scales and in diverse biological structures.

---

## [Referee Report · Reviewer #1 (Public Review)]

The present paper presents a new, simple, and cost-effective technique for multimodal EM imaging that combines the strengths of volume scanning electron microscopy (SEM) and electron microscopic tomography. The novel ATUM-Tomo approach enables the consecutive inspection of selected areas of interest by correlated serial SEM and TEM, optionally in combination with CLEM, as demonstrated. The most important finding of ATUM-Tomo and particularly correlative ATUM-Tomo is that it can bridge several scales from the cellular to the high-resolution subcellular scale, from the micrometer to low nanometer resolution, which is particularly important for the ultrastructural analysis of biological regions of interest as demonstrated here by focal pathologies or rare cellular and subcellular structures. Both imaging modalities are non-destructive, thus allowing re-imaging and hierarchical imaging at the SEM and TEM levels, which is particularly important for precious samples, such as human biopsies or specimens from complex CLEM experiments. The paper demonstrates that the new approach is very helpful in analyses of pathologically altered brains, including humans brain tissue samples, that require high-resolution SEM and TEM in combination with immunohistochemistry for analysis. Even the combination with tracers would be possible. In sum, ATUM-Tomo opens up new possibilities in multimodal volume EM imaging for diverse biological areas of research.

Strengths

This paper is a very nice piece of work. It combines modern, high-end, state-of-the-art technology that allows to investigate diverse biological questions in different fields and at multiple scales. The paper is clear and well-written. It is accompanied by excellent figures, supplementals, and colored 3D-reconstructions that make it easy for the reader to follow the experimental procedure and the scientific context alike.

Weaknesses

There is a bit of an imbalance between the description of the state-of-the-art methodology and the scientific context. The discussion of the latter could be expanded.

---

## [Referee Report · Reviewer #2 (Public Review)]

Kislinger et al. present a method permitting a targeted, multi-scale ultrastructural imaging approach to bridge the resolution gap between large-scale scanning electron microscopy (SEM) and transmission electron microscopy (TEM). The key methodological development consists of an approach to recover sections of resin-embedded material produced by Automated Tape Collecting Ultramicrotomy (ATUM), thereby permitting regions of interest identified by serial section SEM (ATUM-SEM) screening to be subsequently re-examined at higher resolution by TEM tomography (ATUM-Tomo). The study shows that both formvar and permanent marker coatings are in principle compatible with solvent-based release of pre-screened sections from ATUM tape (carbon nanotubule or Kapton tape). However, a comparative analysis of potential limitations and artifacts introduced by these respective coatings revealed permanent marker to provide a superior coating; permanent marker coatings are more easily and reliably applied to tape with only minor contaminants affecting the recovered section-tape interface with negligible influence on tomogram interpretation. Convincing proof-of-principle is provided by integrating this novel ATUMTomo technique into a technically impressive correlated light and electron microscopy (CLEM) approach specifically tailored to investigate ultrastructural manifestations of trauma-induced changes in blood-brain barrier permeability.

Strengths

Schematics and figures are very well-constructed, illustrating the workflow in a logical and easily interpretable manner. Light and electron microscope image data are of excellent quality, and the efficacy of the ATUM-Tomo approach is evidenced by a qualitative assessment of ATUM-SEM performance using coated tape variants and a convincing correlation between scanning and transmission electron microscopy imaging modalities. Potential ultrastructural artifacts induced via solvent exposure and any subsequent mechanical stress incurred during section detachment were thoroughly and systematically investigated using appropriate methods and reported with commendable transparency. In summary, the presented data convincingly support the claims of the study. A major strength of this work includes its general applicability to a broad range of biological questions and ultrastructural targets demanding resolutions exceeding that obtained via serial section and destructive block-face imaging approaches alone. The level of methodological detail provided is sufficient for replication of the ATUM-Tomo technique in other laboratories. Consequently, this relatively simple and cost-effective technique is widely adoptable by electron microscopy laboratories, and its integration into existing ATUM-SEM workflows supports a versatile and non-destructive imaging regime enabling high-resolution details of targeted structures to be interpreted within anatomical and subcellular contexts.

Weaknesses

I find no significant weaknesses in the current version of the manuscript.

---

## [Author Response]

The following is the authors’ response to the original reviews.

Reviewer #1: The authors may consider moving the supplemental figures into the main body of the paper since they finally would end up with a total of eight figures.

As we added two more supplementary figures, we left them separated from the main part of the manuscript in the supplement. All of them describe important experimental details but we believe that it is easier to follow if there is a focus on the key results.

Reviewer #1: In general, the methods and techniques used here are beside some required but important additions described in sufficient detail.Reviewer #2: Given the identified importance of glow-discharge treatment of precoated tape to the flat deposition of sections during ATUM, a corresponding schematic or appropriate reference(s) providing more information about the custom-built tape plasma device would likely be a prerequisite for effective reproduction of this technique in other laboratories.

Thank you for the valuable comments on the missing experimental details, which could affect the ease of establishing ATUM-Tomo in other labs. We will clearly highlight the ATUM-Tomo-specific vs. some general EM processing steps of the workflow in the proposed way. A detailed description of the custom-built tape plasma device will be added to the methods section. In addition, we will reference more explicitly our published protocols, which describe the standard electron microscopy embedding steps in great detail (Kislinger et al., STAR protocols, 2020; Kislinger et al., Meth Cell Biol, 2023).

Reviewer #1: Concerning the results section: In my opinion, the results section is a bit unbalanced. There is a mismatch between the detailed description of the methodology (experimental approach) and the scientific findings of the paper. The reviewer can see the enormous methodological impact of the paper, which on the other hand is the major drawback of the paper. To my opinion, the authors should also give a more detailed description of their scientific results.Concerning the discussion: It would have been nice to give a perspective to which the described methodology can be used not only to describe diverse biological aspects that can be addressed and answered by this experimental approach. For example, how could this method be used to address various questions about the normal and pathologically altered brain?In my opinion, the paper has one major drawback which is that it is more methodologically based although the authors included a scientific application of the method. The question here is to balance the methodology vs. the scientific achievement of this paper, a decision hard to take. In other words, one could recommend this paper to more methodologically based journals, for example, Nature Methods.

Balancing the technological and biological parts is indeed a difficult issue. We agree that this manuscript mainly describes a technical advancement and demonstrates its power to answer previously unsolved scientific questions. We exemplify this in our model system, neuropathology of the blood-brain barrier. The biological impact of ATUM-SEM has been described in detail in Khalin et al., Small, 2022, and is referenced accordingly. Here we describe how ATUM-Tomo can be applied to reveal biological insights exceeding the capabilities of ATUM-SEM and other volume electron microscopy techniques. However, the description of the methodological development outweighs by far the one of the biological details. We consider eLife‘s Tools and Resources (which, in our view, is in scope similar to Nat Methods) an ideal format for this technically focused manuscript while targeting eLife’s readership with diverse biological fields of interest for potential applications of the method. We suggested the application in connectomics (for chemical synapses), the study of endocytosis and the detection of virus particles in the discussion. Hopefully, this accommodates the Reviewer’s concern that having only a single application might seem arbitrary or even suggest a very narrow utility of the technique.

“While we demonstrate a neuropathology-related application, further biological targets that require high-resolution isotropic voxels and the spatial orientation within a larger ultrastructural context can potentially be studied by ATUM-Tomo. This includes the detection of gap junctions for connectomics or for the study of long-range projections (Holler et al., 2021) and the subcellular location of virus particles (Wu et al., 2022, Roingeard, 2008, Pelchen-Matthews and Marsh, 2007). Thus, ATUM-Tomo opens up new avenues for multimodal volume EM imaging of diverse biological research areas.”

Reviewer #2: Is the separation of sections from permanent marker-treated tape sensitive to the time interval between deposition/SEM imaging and acetone treatment?

Thank you for pointing out this important methodological aspect. We have not systematically investigated whether there is a critical time window between microtomy, SEM, and detachment. From the samples generated for this study, we assessed the importance of timing in retrospect:

“The sections could be recovered even four months after collection and nine months after SEM imaging.”

Reviewer #2: To what extent is slice detachment from permanent marker-treated tape resin-dependent [i.e. has ATUM-Tomo been tested on resin compositions beyond LX112 (LADD)]?

We appreciate this comment addressing the broader technical applicability of ATUM-Tomo. We tested the general workflow with tissue embedded in other commonly used resin types (epon, durcupan).

Reviewer #2: Minor corrections to the text and figures.Line 83: (Khalin et al., 2022) should read (Khalin et al., 2022)Line 86 : 30nm should read 30 nmLine 139: "...morphological normal tight junctions..." should read "...morphologically normal tight junctions..."Line 283: "....despite glutaraldehyde fixation, a prerequisite for optimal ultrastructural preservation...".Line 295: "In contrast, our CLEM approach provides high ultrastructural quality by optimal chemical fixation".The concepts of optimal preservation and optimal fixation are arguably context- and application-dependent. These statements should be toned down or their context explicitly stated.

Thank you for the detailed corrections. We have applied them accordingly.